# Cracking, Microstructure and Tribological Properties of Laser Formed and Remelted K417G Ni-Based Superalloy

**Shuai Liu [1], Haixin Yu [1], Yang Wang [1], Xue Zhang [1], Jinguo Li [2], Suiyuan Chen [1] and Changsheng Liu [1,\***

[1]   Key Laboratory for Anisotropy and Texture of Materials Ministry of Education, School of Materials Science and Engineering, Northeastern University, Shenyang 110819, China; 13840524163@163.com (S.L.); 13840362963@163.com (H.Y.); 15040369625@163.com (Y.W.); 18804025352@163.com (X.Z.); chensy@atm.neu.edu.cn (S.C.)

[2]   Institute of Metal Research, Chinese Academy of Sciences, Shenyang 110006, China; jgli@imr.ac.cn

\*   Correspondence: csliu@mail.neu.edu.cn; Tel.: +86-24-83691579

**Abstract:** The K417G Ni-based superalloy is widely used in aeroengine turbine blades for its excellent properties. However, the turbine blade root with fir tree geometry experiences early failure frequently, because of the wear problems occurring in the working process. Laser forming repairing (LFR) is a promising technique to repair these damaged blades. Unfortunately, the laser formed Ni-based superalloys with high content of (Al + Ti) have a high cracking sensitivity. In this paper, the crack characterization of the laser forming repaired (LFRed) K417G—the microstructure, microhardness, and tribological properties of the coating before and after laser remelting—is presented. The results show that the microstructure of as-deposited K417G consists of $\gamma$ phase, $\gamma'$ precipitated phase, $\gamma + \gamma'$ eutectic, and carbide. Cracking mechanisms including solidification cracking, liquation cracking, and ductility dip cracking are proposed based on the composition of K417G and processing characteristics to explain the cracking behavior of the K417G superalloy during LFR. After laser remelting, the microstructure of the coating was refined, and the microhardness and tribological properties was improved. Laser remelting can decrease the size of the cracks in the LFRed K417G, but not the number of cracks. Therefore, laser remelting can be applied as an effective method for strengthening coatings and as an auxiliary method for controlling cracking.

**Keywords:** K417G Ni-based superalloy; laser forming repairing; laser remelting; microstructure; cracking behavior; tribology

## 1. Introduction

Because of the objective reality of enhancing aeroengine performance, blades must work in an environment of high temperature, overloading, and high frequency vibration [1]. The Ni-based superalloy K417G containing a high content of Al + Ti (>7.0 wt.%) is widely used in aeroengine turbine blades for its excellent high-temperature properties and relatively low fabricating cost [2–5]. However, the turbine blade root with fir tree geometry experiences early failure frequently, because of the wear problems occurring in the working process [1]. From an efficiency and economic point of view, it is more appealing and significant to repair the defected or damaged blades instead of replacing them with new ones. Laser forming repairing (LFR), also called laser cladding, is a kind of metal additive manufacturing technology. It can be applied to form a repaired coating that recovers complex or various defected parts up to certain degree and to form a metallurgical bond between substrate and coating, without degrading the inherent service properties of the parts [6–8].

Unfortunately, cracking behavior frequently occurs when laser rapid forming technology is used to manufacture nickel-based superalloy containing a high content of Al + Ti (>7.0 wt.%), which is the most harmful defect that seriously affects the reliability of an aeroengine. Consequently, it is significant to explore the cracking mechanism of the nickel-based superalloy and to seek methods of controlling the cracking behavior. Ojo and Chaturvedi deemed that the constitutional liquation of $\gamma'$ phase was the main factor in liquation cracks in the Inconel 738 welding process [9,10]. Similarly, Li et al. repaired the damaged K465 superalloy turbine blades by LFR and came to the conclusion that the constitutional liquation of $\gamma'$ phase resulted in the formation of liquation films during the K465 repairing process [11]. Tancret et al. computed the liquated $\gamma'$ phase temperature of Inconel 738 by Thermo-Calc software and analyzed the relationship between the heating rate and the dissolution behavior of $\gamma'$ phase [12]. Zhou and Li et al. pointed out that low melting point phases at grain boundaries such as $\gamma + \gamma'$ eutectics and carbides were the main factors in the cracking behavior in the K3 nickel-based superalloy during laser cladding [13,14]. Yang et al. proposed three cracking mechanisms based on the composition of Rene 104 and processing characteristics to explain the cracking behavior of the Rene 104 superalloy during direct laser fabrication [15].

Although a few published papers have investigated different mechanisms of nickel-based superalloys during the laser forming process, limited works have been carried out on the K417G superalloy. So far, the microstructure and properties of laser formed K417G, the cracking mechanism, and control methods of controlling the cracking behavior are still unclear. Moreover, except for component factors and processing parameters, another approach to affect the cracking behavior is post treatment. Laser remelting is considered as an effective post treatment to improve the quality and properties of coatings. It has been extensively adopted to prepare coatings with a dense structure and excellent properties [16–20]. Therefore, laser remelting can be attempted to decrease or even eliminate the cracks in the laser forming repaired (LFRed) K417G. In this paper, the microstructure observation and crack analysis of the LFRed K417G are presented. Then, cracking mechanisms are proposed, taking the chemical composition of the K417G superalloy and laser processing characteristics of LFR into account. Finally, the effects on microstructure, cracking behavior, hardness, and tribological properties of the LFRed K417G after laser remelting are investigated.

## 2. Materials and Methods

The substrate used in this experiment was the as-cast K417G superalloy with dimensions of ø 25 mm × 8 mm. The K417G spherical powder was supplied by Institute of Metal Research of Chinese Academy of Sciences (Shenyang, China) and refined by the gas atomization method. The particle size of powder was 50–150 μm and its specific elemental composition was 0.14C, 9.84Cr, 6.37Al, 4.79Ti, 11.4Co, 3.18Mo, 2.80Fe, and balance Ni (wt.%). The laser equipment used in this experiment was a laser direct deposition forming system (Key Laboratory for Anisotropy and Texture of Materials, Ministry of Education, Shenyang, China), which mainly consisted of a YAG-1000W fiber laser, protective atmosphere device, self-designed coaxial ring powder feeding device, circulating cooling device, and computer system for forming control. In the laser repairing process, the positive defocusing modes were adopted to obtain a smaller dilution ratio and a higher cladding efficiency, the defocus amount was 4 mm, and the spot diameter was 1.8 mm. High purity argon (99.99%) was used as the bath protection gas to prevent oxidation, and the shielding gas flow rate was 7 L/min. According to the relevant studies [21–23], the process parameters of LFR were optimized. The LFR process parameters of each layer are shown in Table 1 and the schematic of laser scanning path is shown in Figure 1. The deposited coating was ten layers in order to ensure adequate thickness. The laser remelting process parameters were the same as the deposition process, except that no powder was fed. When the deposition of each layer was completed, the powder feeding was stopped and the surface of deposited layer was laser scanned again along the original path at that height. Each layer was remelted once. Then, the focus was raised by 0.4 mm to continue the process of deposition and remelting of the next layer. Because there were ten layers of the coating, the total of remelting times was ten. The remelting

process did not bring about a great change in thickness of each deposited layer. Before and after remelting, the thickness of each layer was about 0.4 mm. The thicknesses of two ten-layer coatings were similar, both of which were 3.8 mm.

**Table 1.** Main process parameters of laser forming repairing.

| Laser Power (W) | Scanning Speed (mm/s) | Powder Feeding Amount (g/min) | Powder Flow (L/min) | Overlap Rate (%) | Z-Axis Lift (mm) | Interlayer Cooling Time (min) |
|---|---|---|---|---|---|---|
| 600 | 5.4 | 5 | 3.5 | 40 | 0.4 | 10 |

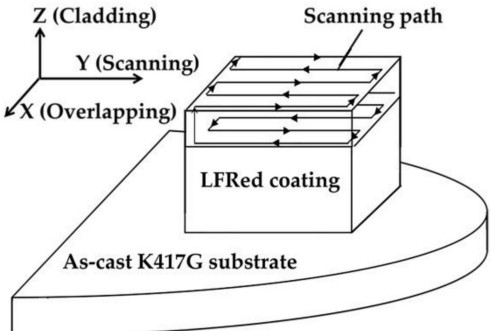

**Figure 1.** The schematic of laser scanning path. LFR—laser forming repairing.

After preparing, the LFRed coatings were cut into reasonable size blocks together with the substrate by numerically controlled wire-cutting. After being mechanically ground and polished, the sectioned samples were electrolytically etched in 15 g $CrO_3$ + 10 mL $H_2SO_4$ + 150 mL $H_3PO_4$ at 5 V for 35–50 s. Samples were ultrasonically cleaned after corrosion for 10 min, and finally rinsed with absolute ethanol and dried. The phase composition was measured using an X-ray diffractometer (XRD) (Smartlab-9000, Rigaku, Tokyo, Japan) machine and the main operating parameters including a 40 kV voltage, 250 mA current, Cu K$\alpha$ radiation, 0.02° angle step-length, and 5°/min scanning rate. Scanning electron microscope (SEM) (JSM-6510A, JEOL, Tokyo, Japan) and its own energy spectrum analyzer (EDS) were used to observe the microstructure and analyze micro-area composition and wear surface of samples. The dendritic spacing and crack size were measured by Image-Pro Plus 6.0 image analysis software (Image-Pro Plus software, Media Cybernetics, Bethesda, MD, USA). The microhardness was measured by a digital micro vickers hardness tester (401MVD, Wolpert Wilson, Norwood, MA, USA). The test areas were regions from the top of the coating to the substrate in the longitudinal section of Y–Z. A load of 200 g was applied, and the holding time was 10 s. The microhardness of each sample was measured three times, and the average of the three results was taken as the microhardness of the sample. The tribological properties of samples were evaluated using Universal friction and wear tester (Nanovea, Irvine, CA, USA) and its own 3D contact surface profiler. The samples with dimensions of ø 15 mm × 10 mm were prepared to conduct the wear experiment at room temperature. The radius of the wear tracks was set to 3 mm using φ6 mm $Si_3N_4$ balls as a counterpart. The measurements were implemented for a sliding length of 54 m with a speed of 15 mm/s, a load of 10 N, and a relative humidity of 60% ± 3%.

## 3. Results and Discussion

### 3.1. Microstructure and Main Phases of LFRed K417G Superalloy

Samples were examined in order to determine the phase composition using an X-ray diffraction analyzer (XRD). Figure 2 is an XRD diffraction pattern of the LFRed K417G. It can be seen that the coating mainly contains $\gamma$ solid solution, Al0.5CNi3Ti0.5 carbide, and $\gamma'$-Ni3(Al,Ti) strengthening precipitation phase. Figure 3 shows the SEM images and the EDS analysis results of line scanning on the X–Y section. The results indicate that Ti and Mo elements are significantly segregated between

the dendrites, while Cr and Co are evenly distributed in the dendrites. Figure 4 shows the typical microstructure of the as-deposited K417G on the X–Z section and the X–Y section. The darker areas in Figure 4a are dendrites, while the brighter areas are interdendritic zones, the measured dendrite spacing is 10–18 μm. There are mainly three types of precipitates between dendrites, namely, finely distributed dot-like precipitates, white block-shaped precipitates, and gray-white fishbone-like tissue. The proportion of white block-shaped precipitates and gray-white fishbone-like tissue is higher, occupying half of the interdendritic zones and distributing nonuniformly.

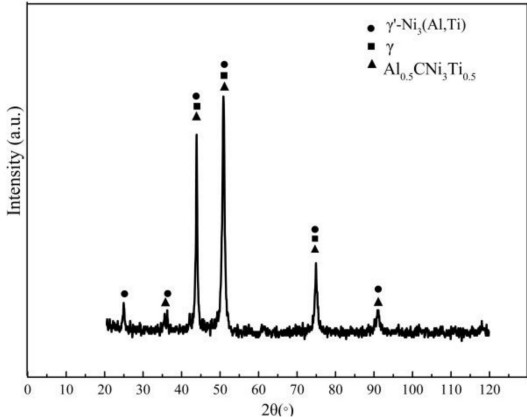

**Figure 2.** X-ray diffractometer (XRD) diffraction pattern of the laser forming repaired (LFRed) K417G.

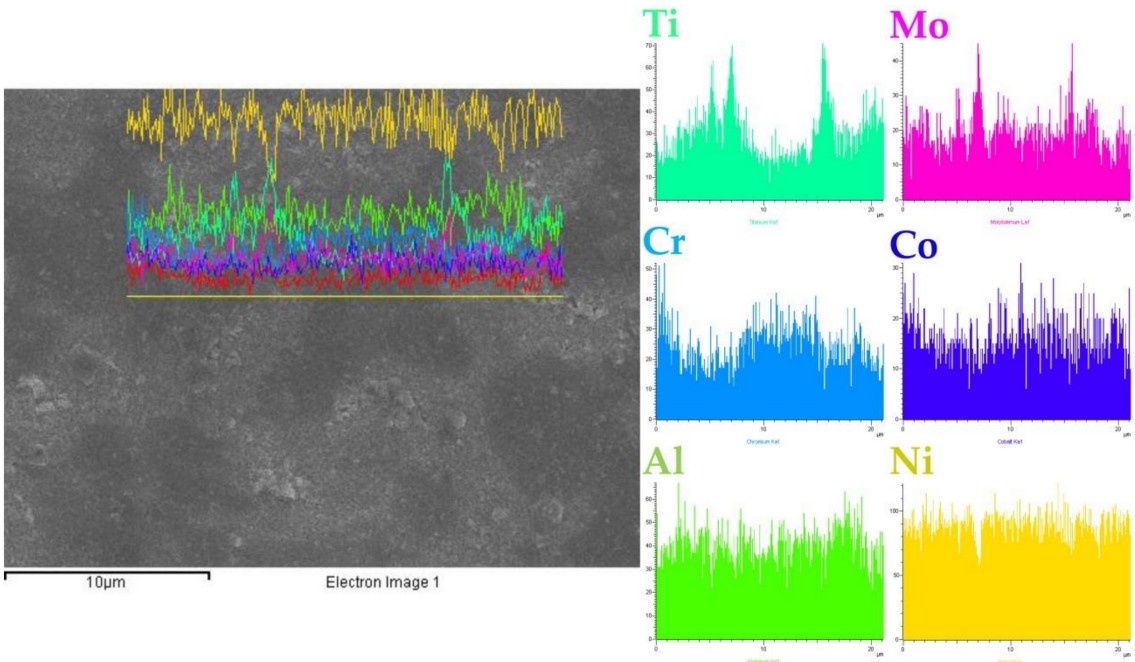

**Figure 3.** Energy dispersive spectrometer (EDS) results of line scanning on the X–Y section.

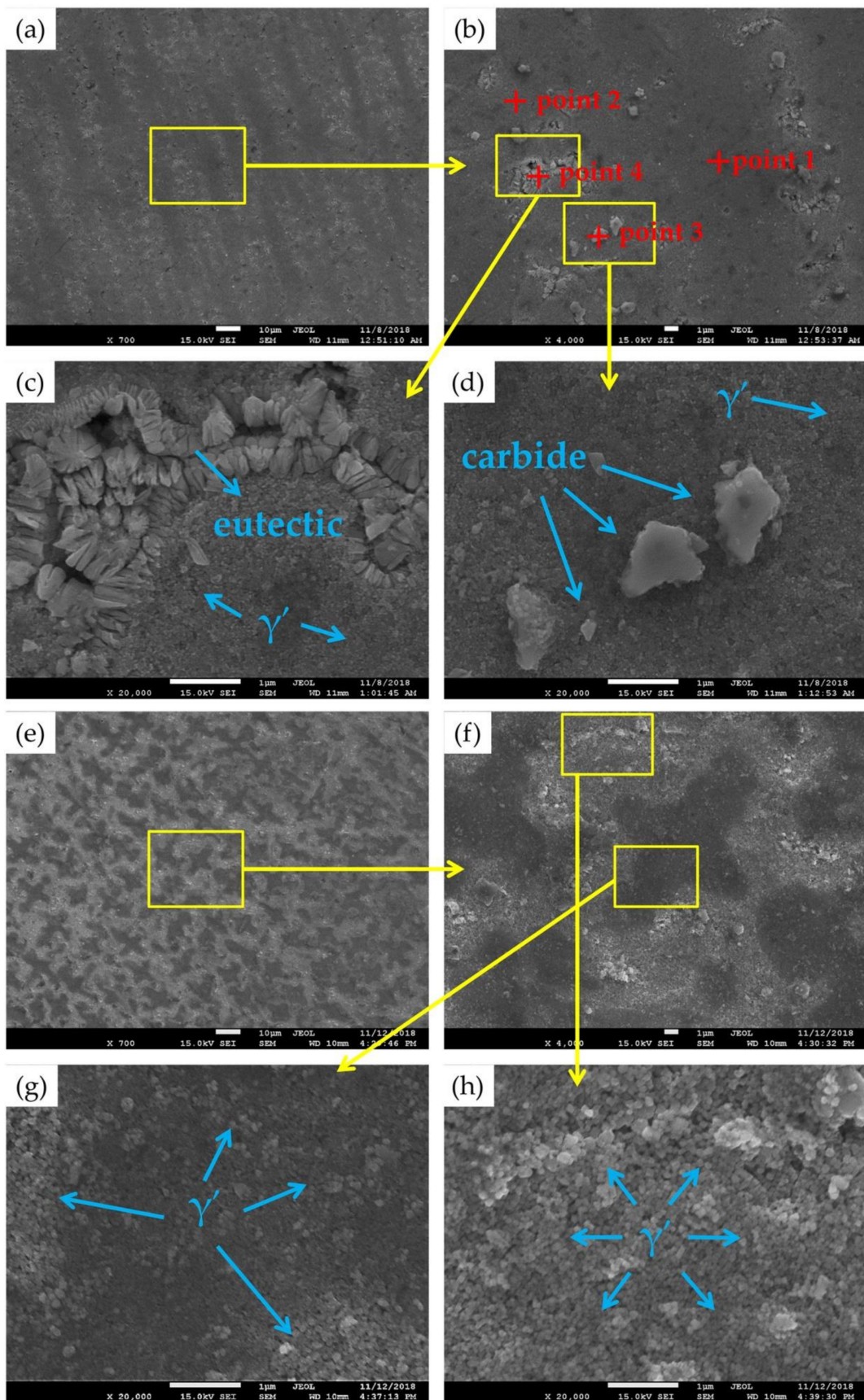

**Figure 4.** Typical microstructure on (**a**–**d**) the X–Z section and (**e**–**h**) the X–Y section of the as-deposited K417G.

In order to further analyze the microstructure and phase composition, energy dispersive spectrometer (EDS) of point scanning is performed on typical locations on the microstructure. As shown in Table 2, the results indicate that the composition of the dark-gray zone in dendrites (point 1 in Figure 4b) is similar to that of the original powder, while the contents of Al, Ti, and Mo are lower than the original powder. This is because during the solidification process, because the laser deposition is a near-rapid cooling process, the Al, Ti, and Mo element will be partly segregated in the remaining liquid phase during the non-equilibrium solidification process. As can be seen from Figure 4g, there are also dot-like precipitates in the dark-gray zone, which are the $\gamma'$ phases precipitated on the matrix. Therefore, the dark-gray zone (point 1 in Figure 4b) is a two phase $\gamma$ plus $\gamma'$ microstructure. The element distribution of finely distributed dot-like precipitates between the dendrites (point 2 in Figure 4b) is close to that of the matrix, but the content of Al and Ti is increased. Combined with the previous XRD results and morphological comparison of precipitates in some literature about Ni-based superalloys [24,25], it can be concluded that the location of point 2 is also $\gamma$ phase plus $\gamma'$ phase Ni3(Al,Ti); in addition, the aggregation of Ti will make the $\gamma'$ phase coarse. Therefore, the morphology and content of $\gamma'$ phase between the dendrites are relatively different from those in the dendrites, as we can see from Figure 4g,h. The location of point 3 (in Figure 4b) shows the white block-shaped precipitates distributed between dendrites. An elemental analysis shows that the C, Ti, and Mo elements are enriched in the particle precipitated phase compared with the matrix phase. Its morphology is characterized by MC-type carbides, where M represents the metal elements Al, Ti, and Mo. According to previous XRD analysis, it is presumed that the precipitation form of MC should be (Al, Ti, Mo) C. However, considering some types of carbides such as M6C, M23C6, and so on are usually hard to detect in XRD, it is difficult to determine whether other types of carbides are present in as-deposited K417G [25,26]. The content of elemental composition at point 4 is similar to that at point 2. The contents of C and Ti are slightly higher, and the form is clustered or fishbone. This is $\gamma + \gamma'$ eutectic structure. The matrix is first formed during solidification, Al and Ti elements are segregated in the liquid to precipitate and grow the $\gamma'$ phase in advance, and finally the remaining liquid is pushed to the interdendritic position to cause eutectic reaction at a lower temperature; then, the eutectic structure forms [25,27–29].

**Table 2.** Elements' concentration of different test points (mass fraction %).

| Element | C | Al | Ti | Cr | Fe | Ni | Mo | Co |
|---------|------|------|------|------|------|-------|------|-------|
| Point 1 | 1.79 | 4.65 | 1.91 | 9.33 | 7.87 | 62.34 | 2.81 | 9.45 |
| Point 2 | 2.61 | 5.85 | 7.36 | 5.73 | 2.46 | 64.13 | 4.62 | 3.52 |
| Point 3 | 8.72 | 5.56 | 6.24 | 9.23 | 5.62 | 51.23 | 3.20 | 5.72 |
| Point 4 | 3.87 | 3.52 | 6.03 | 7.52 | 3.16 | 61.25 | 3.23 | 5.26 |
| Powders | 0.14 | 6.37 | 4.79 | 9.84 | 2.80 | 61.2 | 3.18 | 11.4 |

*3.2. Cracking Behavior and Mechanisms of LFRed K417G Superalloy*

3.2.1. Crack Observation and Analysis

Unfortunately, severe cracking behavior occurs in the as-deposited K417G samples. Figure 5a shows the cracks on the X–Z section, the existence of cracks with the length in the range of 0.2 to 2 mm in the repair zone (RZ) and the heat affected zone (HAZ) can be found. In addition, the cracks on the X–Z section are approximately parallel to the depositional direction, and have the same orientation as the columnar crystal. This is because the grain boundary is considered as a favorable crack initiation site, so that cracks tend to form along the grain boundary, except for a few initial points in the pores, which is similar to the results presented by Carter et al. [30]. The crack count and crack density, defined as the total number of cracks per unit area by Ghosh and Partha, are measured to estimate the cracking sensitivity [31]. Figure 5b gives the crack count and density of the as-deposited K417G samples. According to the observation of different sections and the statistical results in Figure 5, it can be concluded that compared with most superalloys, the cracking sensitivity of the as-deposited K417G is more serious [9–15,32].

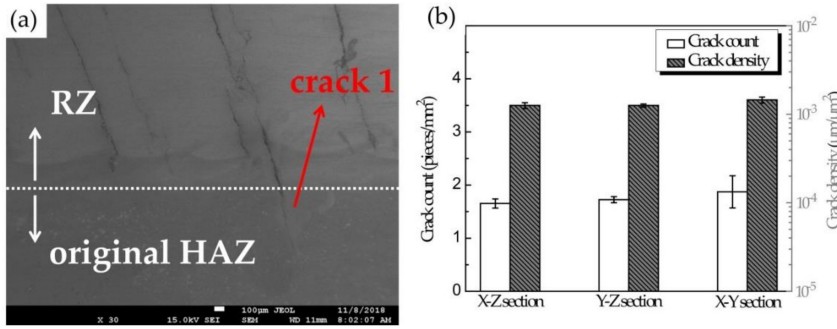

**Figure 5.** (**a**) The cracks on the X–Z section and (**b**) the crack count and density on three sections of as-deposited K417G. RZ—repair zone; HAZ—heat affected zone.

This characteristic of cracking behavior is mainly influenced by the composition of K417G and the processing of LFR. Firstly, K417G has a very high content of Al and Ti (11 wt.%), which are $\gamma'$-Ni3(Al,Ti) forming elements. Consequently, a large amount of $\gamma'$ phase exists in the K417G superalloy. On one hand, these $\gamma'$ phases play a role in strengthening the material. On the other hand, the contraction stresses during the precipitation of $\gamma'$-Ni3(Al,Ti) and the forming of low melting point ($\gamma + \gamma'$) eutectics will be caused by these $\gamma'$ phases, which will dramatically increase the cracking sensitivity of the alloy [25,33]. Secondly, the processing characteristic of LFR is another vital factor that affects the cracking sensitivity. LFR is a process of layer-by-layer superposition. As shown in Figure 6a, the LFRed K417G has an obvious layer-by-layer structure. Also, deposited layers of different heights undergo different thermal cycles. Five types of the thermal cycles with different peak temperatures have been drawn in Figure 6b. In fact, during LFR, when the laser focuses on a certain location, the powder there is heated and rapidly melted. As the laser moves away, the melted material at this location cools rapidly. The material at this location will undergo a complex cycle of repeated heating or even remelting as the next layer of cladding takes place. Any position within the LFRed coating, except for the final solidification areas on the top, will follow this process. The peak temperature of the thermal cycle changes continuously along with the distance from the already solidified position to the molten pool. When the distance is close (for example, the next layer), the reheating temperature of the previous position will be high, whereas when the molten pool is further away, the reheating temperature will be low. As for region A in Figure 6a, when the molten pool is in region A, thermal cycle 1 occurs, whereas when the molten pool is in region B, thermal cycle 2 occurs, and so on. These five cycles correspond to exactly five different distance ranges. This process of repeatedly rapid heating and cooling carries a high risk of cracking [34–37]. It is worth noting that the different regions shown in Figure 6a are distributed in three dimensions. In addition, the regions A, B, C, D, and E are schematic and should actually be much larger than those in Figure 6a.

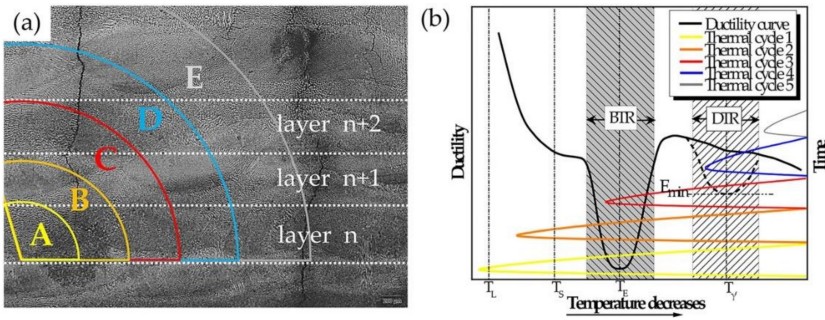

**Figure 6.** (**a**) Microstructure in the middle of the LFRed coating and (**b**) schematic diagram of the combined effect of ductility curve and thermal cycle. BTR—brittleness temperature region; DTR—ductility dip temperature region.

### 3.2.2. Cracking Mechanisms

In order to further explore the influence of composition and process on cracking sensitivity of LFRed K417G, a schematic diagram of the combined effect of ductility curve and the five thermal cycles is drawn. As shown in Figure 6b, the ductility of the material changes as the temperature decreases in the solidification process. There are two regions with obviously low ductility. One is the brittleness temperature region (BTR), which is prone to solidification cracking (SC) and liquation cracking (LC). The other is the ductility dip temperature region (DTR), which is prone to ductility dip cracking (DDC). The two low ductility regions are around the temperature of eutectic reaction temperature ($T_E$) and $\gamma'$ precipitation temperature ($T_{\gamma'}$), respectively, whereas around other temperatures of liquidus temperature ($T_L$) and solidus temperature ($T_S$) show no low ductility regions. In addition, the material will undergo five types of thermal cycles, whose peak temperatures are above $T_L$, above $T_S$, above $T_E$, above $T_{\gamma'}$, and below $T_{\gamma'}$, respectively. The time of each thermal cycle is related to the process of LFR, while the ductility curve is mainly related to the composition of the superalloy. When the thermal cycle of the material is in two regions of BTR and DTR, the cracking behavior will be generated. The cracking behavior in K417G during LFR could be classified into three types. They are solidification cracking, liquation cracking, and ductility dip cracking.

Solidification Cracking

The schematic of the solidification cracking mechanism is shown in Figure 7. When the temperature drops below $T_L$, the primary solid phase of different orientations begins to form, as shown in Figure 7a. When the temperature drops below TS, these different orientations of solids grow alternately to form the grain skeleton. In this case, the residual liquid phase cannot flow freely between the solid dendrites, forming a closed and continuously distributed liquid film. Moreover, the rapid cooling rate results in the non-equilibrium solidification and enrichment of Al and Ti in the interdendritic zones, as shown in Figure 7b. When the content of (Al + Ti) in the liquid phase reaches the critical value, the eutectic reaction of L $\rightarrow$ $\gamma$ + $\gamma'$ will be generated between the dendrites, thus forming the eutectic structure. As shown in Figure 7c, the temperature at this time is TE. Under the action of contraction stress, continuous liquid film ends produce strain concentration, and it is easy to form shrinkage cavity and microcrack in the weak eutectic structure between the dendrites. In the process of continuous solidification (below the temperature of TE), if the interdendritic zone with shrinkage cavity and microcrack is at the grain boundary, the microcrack will propagate along the brittle grain boundary, thus forming the solidification crack. This cracking behavior occurs when the material experiences the thermal cycles of 1 and 2 (the yellow line and orange line in Figure 6b).

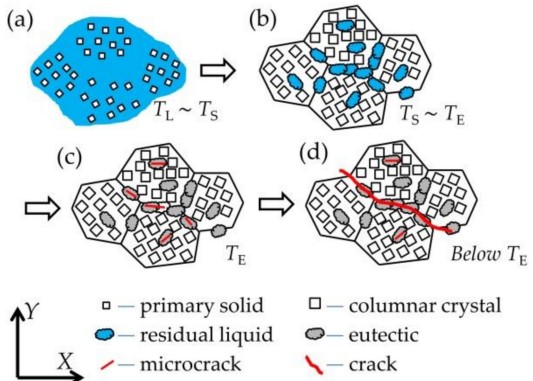

**Figure 7.** Schematic diagram of the formation process of solidification cracking.

In the process of crack growth, the residual stress is gradually released and the energy required for crack growth is gradually reduced. At the same time, as the deposited layer in the extension zone has solidified, the closer to the substrate, the more completely solidified, and the greater the strength

of the intercrystalline bond. When the stress is not enough to break the intercrystalline bond, the crack will be terminated. In addition, as the number of deposited layers increases, the tensile stress in the repaired area gradually decreases and changes to compressive stress, and the crack expansion along the deposited direction towards the top will be impeded. Therefore, most solidification cracks will appear perpendicular to the direction of deposition. Figure 8a,b show images at a high magnification of a crack on the X–Y section. Figure 8c shows the morphology of the smooth area of the crack section, where there are almost no holes in the depth direction, which is determined by the nearly two-dimensional distribution of the interdendritic liquid film. Because the dendrites near the grain boundary remain free in the residual liquid phase and grow without solid obstacle, and the cracking is carried out along the grain boundary, the dendrites near the crack appear as round particles with a smooth surface, as shown in Figure 8d. Consequently, it can be seen that the crack in Figure 8 belongs to a solidification crack.

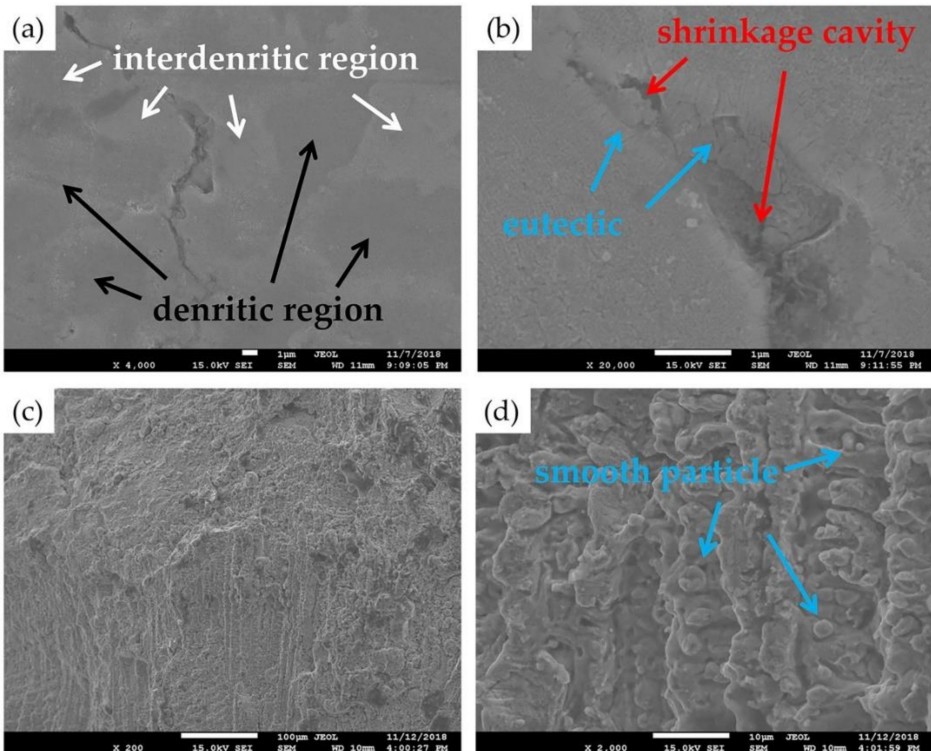

**Figure 8.** Morphology of (**a**,**b**) a solidification crack and (**c**,**d**) its section.

Liquation Cracking

Liquation cracking is the most common form of cracking behavior in superalloys with high (Al + Ti) content. Unlike solidification cracking, the source of liquation cracking is low-melting point eutectics in the heat affected zone (HAZ). These solidified eutectics will be re-melted to form liquid film when the reheated temperature is above TE, which will lead to intergranular cracking under the action of contraction force. This cracking process occurs during the thermal cycles of 3 (the red line Figure 6b). Figure 9 shows the formation process of liquation cracking. It is worth noting that the HAZ does not only exist in the substrate, but also in the previous layer when the laser is focusing on the current layer. In Figure 6a, for example, when the molten pool is in region C, the solidified region A will be a HAZ. Whereas when the molten pool is in region D, the solidified region B will be a HAZ at this time. In other words, there are countless HAZs inside the coating as long as the laser melting is not performed only once. To be exact, the HAZ shown in Figure 5 is the only original HAZ. With the continuous progress of LFR process, HAZs are constantly formed, and the eutectics with low-melting points in HAZs are constantly remelted, thus forming more and more crack sources. Because of the

characteristic of epitaxial growth on the structure during the LFR process, the structure between layers has the genetic characteristic, which easily form columnar crystals with the same orientation. Consequently, the channels through the columnar dendrites between layers are generated. Once the liquefied microcrack is formed at the grain boundary, it will extend along the grain boundary with the accumulation of residual tensile stress and form the macroscopic crack through many deposited layers.

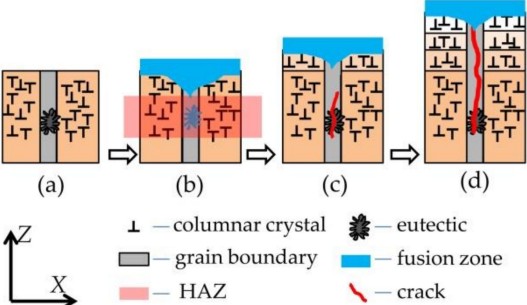

**Figure 9.** Schematic diagram of the formation process of liquation cracking, (**a**) a eutectic in a HAZ; (**b**) the liquefied eutectic; (**c**) the formation of a microcrack; (**d**) the extension of the microcrack.

Liquation cracks usually grow along the direction of dendrite growth, which is similar to an upward growth direction parallel to the deposition direction. As shown in Figure 10a, the extension direction of the crack seems to follow the growth law above. Moreover, it can be seen from Figure 10b that the crack is filled with broken liquid film, and eutectic and coarse $\gamma'$ are distributed around the crack. Figure 10c shows the macroscopic morphology of the crack section. It has the characteristic of intergranular cracking. As shown in Figure 10d, obvious liquation of dendrite protrusions can be observed on the microscopic morphology of the crack section. The crack section takes on the shape of a potato of different sizes, indicating that it is the result of interdendritic liquid film separation and is a typical liquation crack. However, for the judgment of cracking mechanism of other cracks in Figure 5a, it is not sufficient to observe only the growth direction. Because of the complex solidification process of melting, remelting, partial remelting, cyclic annealing, and countless HAZ within the LFRed coating, it is difficult to identify whether the cracking behavior belongs to the liquation cracking or solidification cracking [38,39]. It can at least be determined that crack 1 in Figure 5a belongs to the liquation crack, as it exists in the original HAZ and is similar to an upward growth direction parallel to deposition.

Ductility Dip Cracking

When the material is experiencing thermal cycle 4 (the blue line Figure 6b), it will suffer the effect of continuously growing solid-phase shrinkage stress. No liquid phase exists in this process, and the deformation mode mainly depends on the vacancy diffusion or the dislocation climb along the grain boundary. When the processes of diffusion and climb are occurring, the dislocation will meet obstacles. If the ductility of the material is poor, cracking will occur due to the strain concentration. At this moment, the crack belongs to ductility dip crack. As these obstacles can be the vertex where the three grains intersect, or the carbides on the grain boundary, ductility dip cracking is generally generated by two types, as shown in Figure 11. The straight grain boundaries promote grain boundary sliding to form large voids and corresponding strain concentration at a vertex where the three grains intersect. These voids at the vertex eventually develop into a crack, which is type 1 in Figure 11. Type 2 in Figure 11 is mainly related to the carbides. In fact, the carbides here play a double role in the effect on the sensitivity of ductility dip cracking. On the one hand, the carbides lock the grain boundaries and reduce the grain boundary sliding, thus reducing the strain concentration at the vertex of three adjacent grains. The carbides, on the other hand, lock the grain boundaries but accumulate strain and voids around the carbides themselves, which may lead to the formation of cracks [40]. The effect of carbides on the sensitivity of ductility dip cracking depends on their type, size, and distribution.

The situation is very complicated. Emin in Figure 6b is the critical strain value of ductility dip cracking in the DTR, and its value can be used as an indicator to determine the sensitivity of ductility dip cracking of the material. When the carbides play a role in inhibiting ductility dip cracking, Emin will be slightly below the ductile curve, at which time the width of the DTR is very narrow or even does not exist.

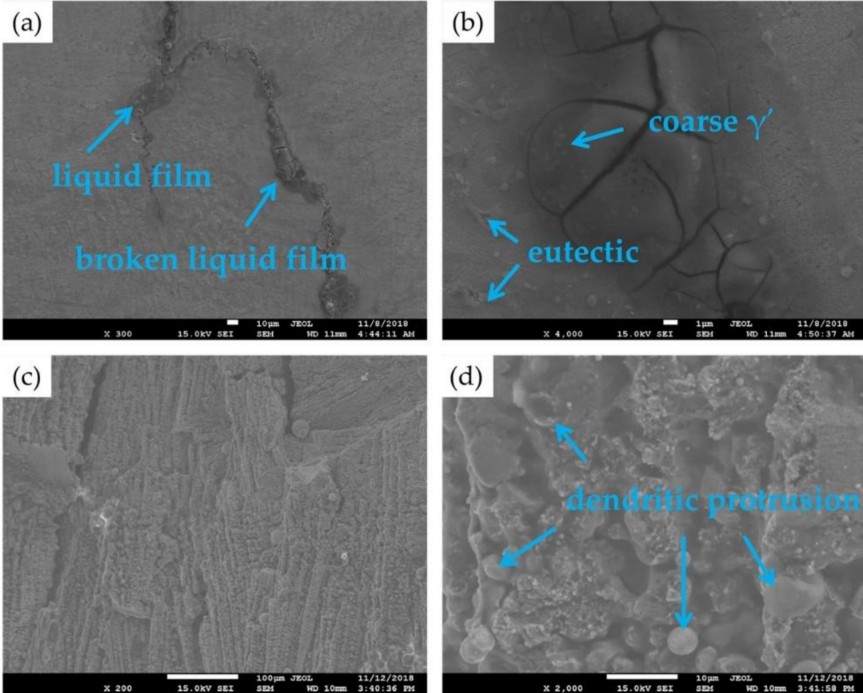

**Figure 10.** Morphology of (**a**,**b**) a liquation crack and (**c**,**d**) its section.

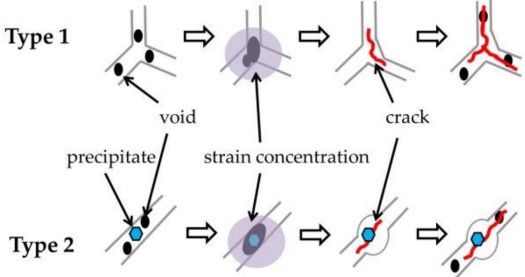

**Figure 11.** Schematic diagram of the formation process of ductility dip cracking.

The crack shown in Figure 12a passes through the vertex of the three adjacent grains and extends along the grain boundaries of these three grains. Moreover, the section in Figure 12b shows that no liquid film exists at the grain boundary of this crack. Consequently, it indicates that the crack should be a ductility dip crack formed in type 1 in Figure 11. However, because it is difficult to find ductility dip cracks formed in type 2, and if solidification cracking and grain boundary liquation cracking have occurred, the cracks expansion at this temperature will be further intensified or connected with ductility dip cracks. Therefore, the effect of the carbides on the sensitivity of ductility dip cracking in LFRed K417G is still unclear.

In summary, it can be determined that the cracking mechanism of cracks in LFRed K417G is solidification cracking, liquation cracking, and ductility dip cracking, respectively. However, the judgment of cracking mechanism of a crack requires comprehensive evidence, which should be combined with observation of macroscopic and microscopic morphology, determination of composition, and analysis of fracture, among others. Moreover, the judgment may still not be

completely accurate, because sometimes it may be a combination of multiple cracking mechanisms. Fortunately, all cracking behaviors are related to the composition of the material and the process of LFR. Therefore, adjusting composition, optimizing process parameters, and post treatment can be utilized to control or eliminate cracking behavior.

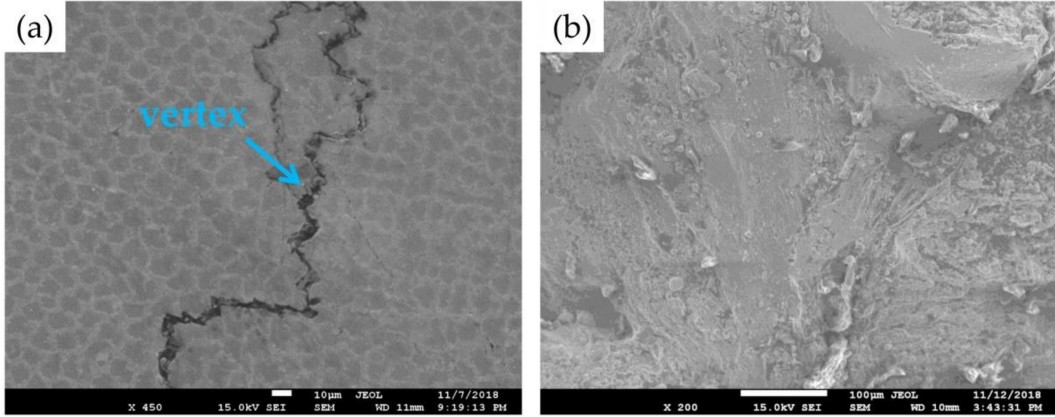

**Figure 12.** Morphology of (**a**) a ductility dip crack and (**b**) its section.

*3.3. Effect of Laser Remelting Process on Microstructure and Cracking Behavior*

3.3.1. Microstructure after Remelting

Laser remelting is considered as an effective post treatment to improve the quality and properties of LFRed coating. As shown in Figure 13, the surface quality of the coating after laser remelting is obviously superior to that before remelting. Laser remelting is also the process of reheating and solidification of the material. No new alloy powder is added during this process. Sharp points and small metal particles can absorb laser energy and remelt. It can be seen that a laser remelting process after laser deposition can make the surface smooth.

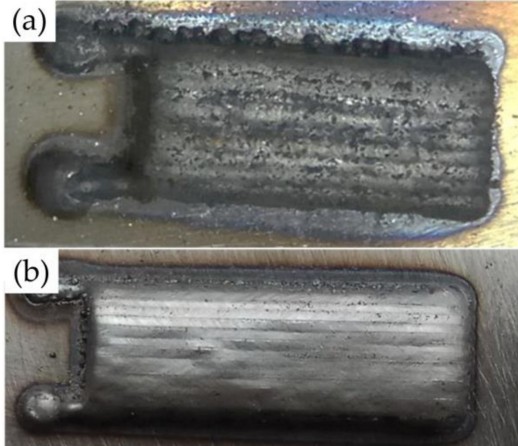

**Figure 13.** Surface morphology of the LFRed coating on K417G substrate (**a**) before and (**b**) after laser remelting.

Figure 14 shows the microstructure of LFRed K417G coating after laser remelting. The microstructure is still mainly composed of $\gamma$ phase, $\gamma'$ precipitated phase, $\gamma + \gamma'$ eutectic, and carbide. After remelting, the microstructure is obviously refined, and the size of the precipitate is reduced. It can be seen from the comparison between the line scan results in Figures 3 and 15 that the element segregation is somewhat reduced and the element distribution is more uniform.

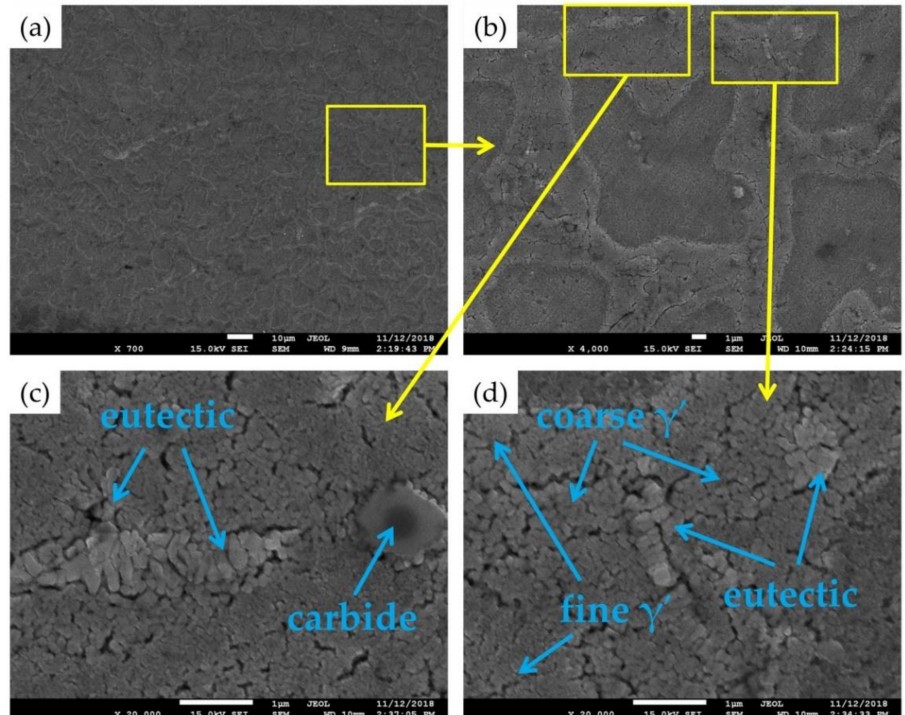

**Figure 14.** Typical microstructure on the X–Z section of the as-remelted LFRed K417G.

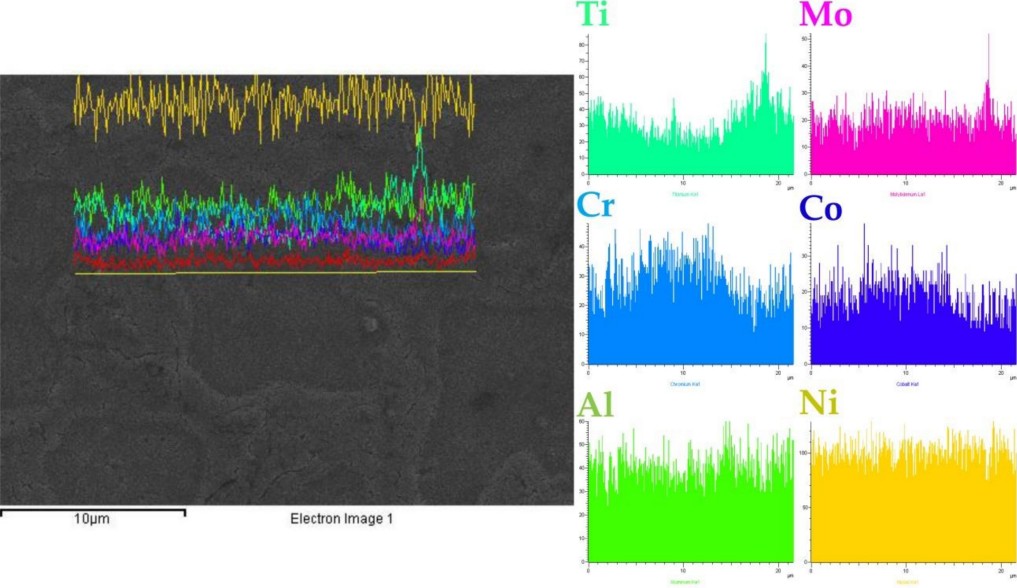

**Figure 15.** Energy dispersive spectrometer (EDS) results of line scanning in the local zone of Figure 14b.

### 3.3.2. Cracking Behavior after Remelting

Figure 16 shows the crack number and crack density of the three sections in the as-remelted K417G coating. According to the statistic of the crack count, the number of cracks in the coating after remelting does not decrease, but increases slightly, which is an undesirable phenomenon. However, the crack density decreases significantly, which is already less than $10^{-3}$ µm/µm$^2$. This indicates a significant reduction in the size of the cracks in the coating. Combined with the analysis of Section 3.2.2, it can be speculated that laser remelting plays a double role in cracking behavior. On the one hand, because of the effect of rapid heating and rapid solidification in the process of LFR, the material has experienced five types of thermal cycles (in Figure 6b), of which four can generate cracks. Laser remelting causes the material to undergo these thermal cycles again, thus doubling the

susceptibility of cracking. Also, the thermal stress is higher because the heat is input again. On the other hand, laser remelting refines the microstructure, reduces elemental segregation, makes the precipitates more evenly distributed, and creates grains with a more different orientation, which undoubtedly hinder the connection of liquid film, thus hindering the extension of microcracks. For large cracks, especially the liquation cracks that have already formed in the HAZ, laser remelting is difficult to heal, and these cracks become larger as a result of higher thermal stress. As a result of this double action, the cracks continue to expand, the number of medium size cracks decreases, and the number of small size cracks increases.

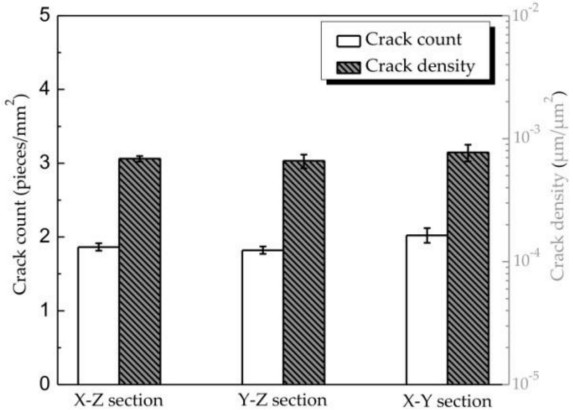

**Figure 16.** Crack count and density on three sections of as-remelted LFRed K417G.

*3.4. Effect of Laser Remelting Process on Microhardness and Tribological Properties*

3.4.1. Microhardness

Figure 17 shows the microhardness of the LFRed K417G coating before and after laser remelting process, from which it can be seen that the as-remelted coating has higher microhardness than that in the as-deposited coating, and the increase is about 50 HV0.2. Moreover, the microhardness of the as-deposited coating changes unsteadily, while the hardness distribution and the dispersion of each measuring point in the as-remelted coating are more uniform. According to the results discussed in Section 3.3, the microstructure after laser remelting is more refined; the distribution of precipitates is more uniform; the size and number of $\gamma + \gamma'$ eutectics, which are harmful to the improvement of hardness, are reduced; and more $\gamma'$ are distributed in dendrites, which has a more significant enhancement effect. These combined effects result in hardness improvement and uniform distribution.

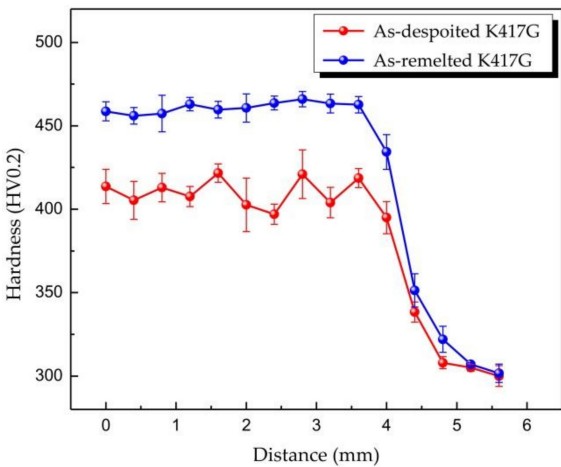

**Figure 17.** Microhardness of the LFRed K417G before and after laser remelting.

3.4.2. Tribological Properties

Friction coefficient evolutions of as-deposited K417G and as-remelted K417G are shown in Figure 18. The raw data have been smoothed (10 points Savitzky–Golay smooth) to facilitate the analysis of the results. At the beginning of the test, the friction coefficients of the two samples reach a high value. This indicates that the friction surface begins to show plastic deformation and adhesion since the beginning of the test, and under the effect of shear stress, the particles begin to peel off on the surface of the material. The average values of friction coefficients in the final 50 m sliding of the as-deposited and as-remelted samples are 0.68 and 0.54, respectively. It is worth noting that this friction coefficient is not a steady value. Because, at the end of the test, the two samples cannot reach the steady state, there are still large fluctuations. In general, the friction coefficient of the as-remelted sample is less than that of the as-deposited sample. Moreover, the friction coefficient of the as-remelted sample shows a downward trend in the late test period, while the as-deposited sample still fluctuates within a larger value.

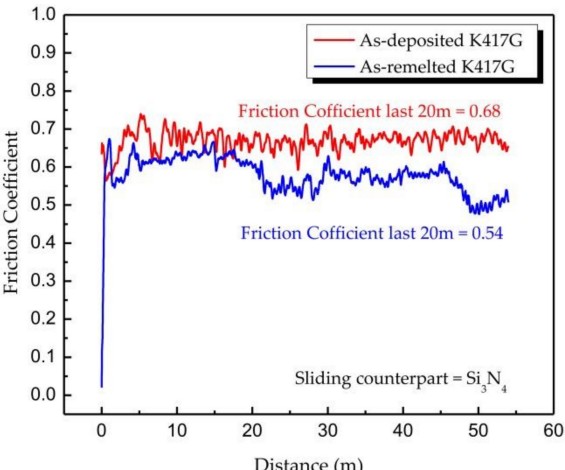

**Figure 18.** The friction coefficient of the LFRed coatings before and after laser remelting process.

As shown in Figure 19, two-dimensional profiles of the wear tracks are obtained at the end of tests. The raw data have been smoothed (10 points Savitzky–Golay smooth) as well. The result clearly shows that after 54 m of sliding, the depth and width of the wear tracks are much larger for the as-deposited K417G than for the as-remelted K417G. The wear rates of the as-deposited K417G and the as-remelted K417G are $5.95 \times 10^{-14}$ and $3.24 \times 10^{-14}$ mm$^3$/N·m, respectively. The wear rate of the coatings agrees well with the expected inverse proportional relation between this property and the hardness of the coatings. All these show that the as-remelted K417G has better tribological properties.

Figure 20 shows the wear surfaces of the as-deposited K417G and the as-remelted K417G. The two wear surfaces are both similar to the form of adhesion wear surface. Large areas of peeling and oxidation can be seen on the wear surfaces. As can be seen from Figure 20c,d, both wear surfaces have deep scratches. These scratches are not similar to those caused by abrasive particles. They are more likely to be caused by scraping of the abrasive chip that has flaked and adhered to the surface. In addition, it can be observed that the cracks on the surface of the as-remelted K417G are separated by the wear surface, while the cracks on the surface of the as-deposited K417G pass through the wear surface, showing that the cracks in the as-remelted K417G are shallower. This confirms, from another aspect, the effect of laser remelting on the reduction of crack size, as discussed in Section 3.3.2.

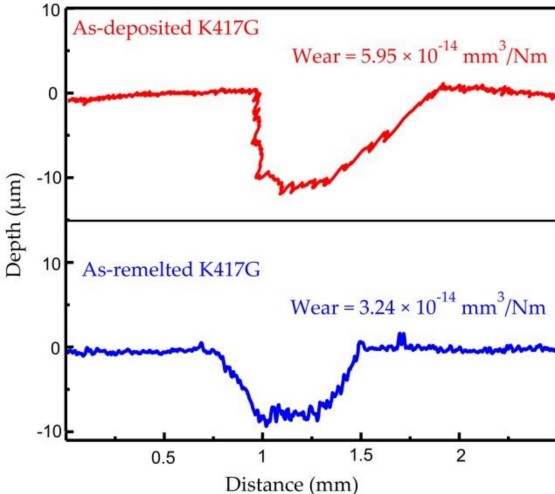

**Figure 19.** The 2D profiles of the wear track of the as-deposited K417G and as-remelted K417G tested against $Si_3N_4$ balls.

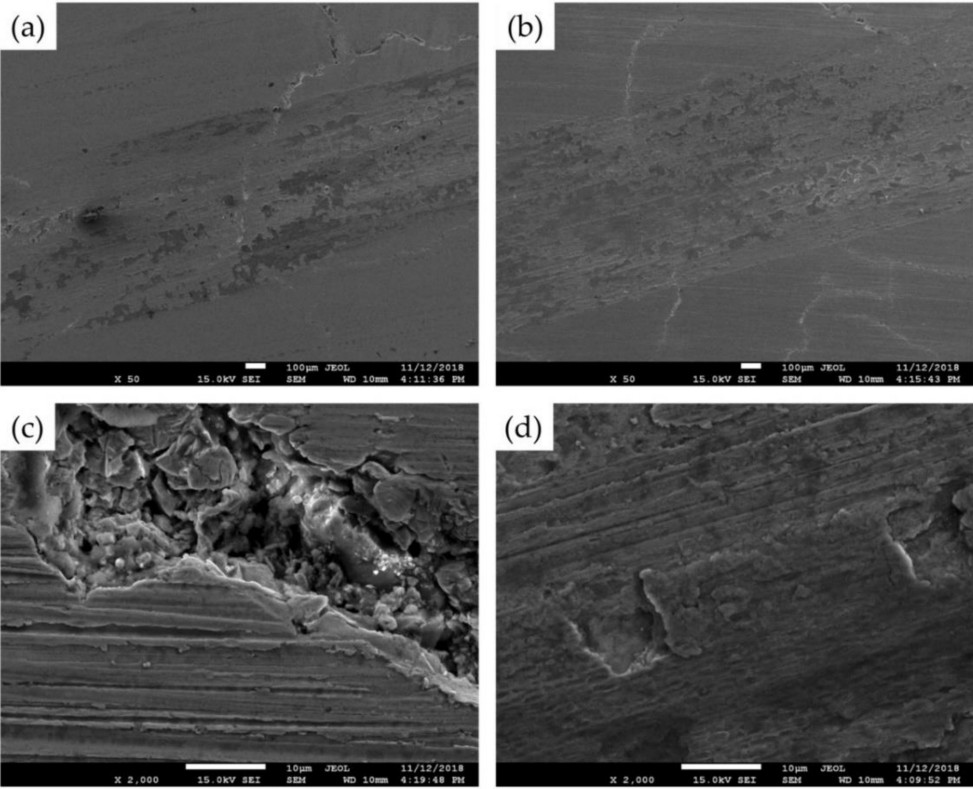

**Figure 20.** Wear surfaces of (**a**,**c**) the as-deposited K417G and (**b**,**d**) the as-remelted K417G.

## 4. Conclusions

The K417G Ni-based superalloy has been prepared on as-cast K417G substrate by the LFR process. The laser remelting process was applied as a post treatment to improve the properties of the LFRed coating. The main conclusions are as follows.

The microstructure of the LFRed K417G consists of $\gamma$ phase, $\gamma'$ precipitated phase, $\gamma + \gamma'$ eutectic, and carbide. The characteristic of cracking behavior is mainly influenced by the composition of K417G and the processing of LFR. Cracking mechanisms of the LFRed K417G include solidification cracking, liquation cracking, and ductility dip cracking.

Laser remelting can decrease the size of the cracks in the LFRed K417G, but not the number of cracks. After laser remelting, the microstructure of the coating was refined, and the element segregation was reduced. The as-remelted coating has higher microhardness, which can reach up to 460 HV0.2 compared with that of the as-deposited coating, and the increase is about 50 HV0.2. Similarly, the as-remelted K417G has a better tribological property than the as-deposited K417G. The wear surfaces are both related to adhesion wear.

Consequently, in the application of LFR technology to repair damaged K417G blades, laser remelting can be applied as an effective method for strengthening coating and as an auxiliary method for controlling cracking. However, cracks still exist. In order to eliminate cracking behavior, more efforts should be committed to component adjustment, process parameter optimization, and other post-treatment studies.

**Author Contributions:** Conceptualization, C.L.; Resources, J.L. and S.C.; Formal Analysis, Y.W. and H.Y.; Data Curation, X.Z. and S.L.; Validation, H.Y. and S.L.; Supervision, C.L.; Writing—Original Draft, S.L.

**Funding:** This research was funded by the Joint Founds of National Natural Science Foundation of China (NSFC)-Liaoning (No. U1508213) and National Key Research Project (No. 2017YFB0305801).

**Acknowledgments:** Bin Zhang and Jing Liang from Northeastern University are gratefully acknowledged for data analysis.

**Conflicts of Interest:** The authors declare no conflict of interest.

## Abbreviations

| | |
|---|---|
| LFR | Laser Forming Repairing |
| LFRed | Laser Forming Repaired |
| HAZ | Heat Affected Zone |
| RZ | Repaired Zone |

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
