# Peer review of "Cracking, Microstructure and Tribological Properties of Laser Formed and Remelted K417G Ni-Based Superalloy"

_coatings, doi:10.3390/coatings9020071_

Round 1
Reviewer 1 Report
Coatings-399338-peer-review
Comments and Suggestions for Authors
The article by Liu et al deals with cracking in Laser Formed and Remelted (LFR) K417G Ni-superalloys. They used atomized powder of the alloy to form the layer deposition on a substrate of the same alloy with the help of a YAG-1000W fiber laser and used XRD, SEM and EDS to characterize the microstructure and crack formation in the deposition layer and after remelting. This study is relevant for gas turbine application were repair of components is increasingly important and economically attractive. The authors appear to have made a systematic study and obtained interesting results, however as a reviewer it is a very difficult task for me to follow their presentation and fully comprehend their work due to the language and the style of writing.
At times the authors have also used unusual terminology not commonly used in superalloy publications – which I presume comes from erroneous translation. For example, they used “rotor blade zigzag crown” I believe to describe the gas turbine blade root with fir tree geometry (?). Another example: “stapling dislocation” probably to describe dislocation pile up. These are only some examples.
The manuscript definitely needs to be improved with respect to the language, as there are also several grammatical errors, as well as some confusing and incomprehensible sentences. Examples:
“However, the aeroengine rotor blade zigzag crown appears early failure frequently, which is because of the wear problems occurring in the working process [1].” (Lines 34-35)
“It can be applied to form a repaired coating which recover complex or various defected parts up to certain degree and to form a metallurgical bond between substrate and coating, without degrading the inherent service properties of the parts [5-7].” (Lines 39-41)
Also there are more serious short comings in the article and I find some descriptions are inadequate / not in enough details for a reader to follow the work. Thus a comprehensive and major revision is essential. I list below some (but not all) points which I find critical:
1. Under 2. Materials and Methods: the authors indicate in Line 86 that “The LFRed and remelted coatings were 10 layers”
However, this description is inadequate as it is not clear if there is one single deposition layer where powder is used and 9 times remelting were done without powder addition, or any other combination of deposition and remelting? The schematic of Fig 1 suggests multiple layer deposition. It must be clearly mentioned in the paper how many layers were deposited during LFR and how many times remelting was done. This is important for understanding the results presented later.
If the layer thickness does not change after remelting, as described – then one can assume that each deposition layer thickness is 3.8 mm divided by the number of deposition layers. The layer thickness is very important and relevant to know in order to follow the later description of microstructure and crack formation. Such details are missing in the text.
2. Figure 2 is an XRD diffraction pattern of the LFRed K417G superalloy coating.
XRD measurements can detect crystal structure of a phase but typically not the composition of the phases. However, in Fig. 2 the authors describe the second phase as Ni-Cr-Co-Mo. In the description in the corresponding text they mention it is the matrix g phase. In Table 2 they also present quantitative measurement of composition from different points in the microstructure and this includes all the elements (C, Al, Ti, Cr, Fe, Ni, Mo, Co) for all points measured by EDS. So it is inadequately described in the paper how the authors concluded that the g phase has only Ni, Cr, Co and Mo. Also it is more appropriate to mark the second phase in Fig 2 as g phase and not Ni-Cr-Co-Mo.
3. In order to further analyze the microstructure and phase composition, …. dark-gray zone (point 1 in Figure 4b) is g matrix phase. (Lines 136-143)
It is concluded here that the EDS measurement at point 1 is from the g matrix. Which means that the microstructure in the dendrite is single phase g matrix? This is surprising as I would expect a two phase g + g’ microstructure. To best resolve this, the authors should present a higher magnification SEM image from this area.
In general, the microstructure images presented in the paper are not always of the right quality / magnification to follow the descriptions of the structure provided. This must be improved.
4. In addition, the cracks are always ….. presented by Carter et al [27]. (Lines 165-169)
It is not clear why crack count and crack density is more in X-Y section than in the other sections? (see Fig. 5b). The authors should comment on this.
Also in Fig 5a, the original HAZ microstructure is hardly visible and almost totally covered by the annotation used. Microstructure of both the HAZ and the interface between HAZ and RZ are very important for readers to follow the results presented in this paper. Therefore, suitable change in Fig. 5a is necessary to take care of this shortcoming. A new image with larger area from the HAZ will be even better.
5. Consequently, a large amount of g' phase exists in the K417G superalloy. (Line 179)
What is the g’ volume fraction? An estimate can be easily obtained from the XRD results.
6. Secondly, the processing characteristic of LFR ….. cladding takes place. (Lines 183-189)
Microstructural study can easily reveal details about the different layers and their thicknesses. It is very important information for this study, but is totally missing. For any understanding of the cracking behavior these microstructural details are absolutely essential.
7. Fig. 6. Some details are not clear.
a. In this diagram - Left to Right is cooling, i.e. temperature decreases (I presume). Such details must be explicitly indicated in the diagram.
b. It must also be mentioned - under what conditions / stages, the different Thermal Cycles (1-5 marked) actually occur during the LFR processing.
Moreover, I fail to understand why both the heating and cooling parts of the Thermal cycles (1-5) are included in the diagram and also why these thermal cycles are symmetrical in heating and cooling? I understand that in the LFR process power is supplied to the system to heat up (and thus is controlled) while during cooling the power is switched off (not controlled), therefore the cycle should be asymmetrical logically.
8. As shown in Figure 6, the ductility of the material changes as the temperature decreases. (Line 197)
What is the basis of this schematic – i.e. is the information based on experimental results or calculations? Also how was ductility measured / calculated at high temperatures?
What are the thermal cycles 1 to 5? Some descriptions are presented in text later, but not fully. It is better to describe the different cycles in the caption of Fig. 6. Also it is better to define the various temperatures (TL, Ts, Tg’, etc.) used in the diagram.
9. …the cracking behavior will be generated. ….. ductility dip cracking in LFRed K417G superalloy. (Lines 207-209)
The whole basis of the conclusion is only a schematic diagram - for which no information is provided - regarding how it was constructed.
10. When the temperature drops below TS, these different orientations of solids grow alternately to form the dendrite skeleton. (Lines 214-215)
Schematic in Fig. 7 shows only a grain structure and there is no correlation revealed with respect to the dendrite structure. On the other hand, the dendrite structure is discussed at length in relation to LFR and cracking.
The List of critical comments is not complete and some more are marked in the text file attached.

Author Response
Dear Reviewer,
I am the author of the manuscript entitled “Cracking, Microstructure and Tribological Properties of Laser Formed and Remelted K417G Ni-based Superalloy”. Before explaining point-by-point the details of the revisions in the manuscript, I must express my gratitude to you. It is a pity that I don't know who you are. If possible, I sincerely hope to have the chance of reading and studying some of your articles. This is because I know from your comments that you are a very professional in this field and a careful person. In fact, we are engaged in the research on the laser-formed blisk made of functionally graded materials. In the process of manufacturing and repairing the blades, we found that cracking behavior frequently occurred when the blade material contains high content of Al + Ti (>7.0 wt. %). In addition, the microstructure and the cracking mechanism of laser-formed K417G are still unclear.
Therefore, we did some correlative research and wrote the paper. Your comments are all valuable and very helpful for revising and improving our paper, as well as the guiding significance to our further research. Besides, you also made me realize my negligence in language learning. Hence I make up my mind to study English well. Thank you again! We have studied your comments carefully and have made correction which we sincerely hope meet with your approval. The main corrections in the paper and the responds to your comments are as follows:
Point 1: Under 2. Materials and Methods: the authors indicate in Line 86 that “The LFRed and remelted coatings were 10 layers”
However, this description is inadequate as it is not clear if there is one single deposition layer where powder is used and 9 times remelting were done without powder addition, or any other combination of deposition and remelting? The schematic of Fig 1 suggests multiple layer deposition. It must be clearly mentioned in the paper how many layers were deposited during LFR and how many times remelting was done. This is important for understanding the results presented later.
If the layer thickness does not change after remelting, as described – then one can assume that each deposition layer thickness is 3.8 mm divided by the number of deposition layers. The layer thickness is very important and relevant to know in order to follow the later description of microstructure and crack formation. Such details are missing in the text.
Response 1: Thank you for pointing this out. The remelting process I described was really not detailed enough. In fact, we only prepared two laser forming repaired (LFRed) coatings in this work. One was the ten-layer as-deposited coating which was done without remelting. The other one was the ten-layer as-remelted coating which consisted of ten remelted layers. Each layer was remelted once. When the deposition of each layer was completed, the powder feeding was stopped and the surface of deposited layer was laser scanned again along the original path at that height.
According to your comment, the detailed description I added is that “The deposited coating was ten layers in order to ensure adequate thickness. The laser remelting process parameters were the same as the deposition process, except that no powder was fed. When the deposition of each layer was completed, the powder feeding was stopped and the surface of deposited layer was laser scanned again along the original path at that height. Each layer was remelted once. Then the focus was raised by 0.4 mm to continue the process of deposition and remelting of the next layer. Because there were ten layers of the coating, the total of remelting times was ten. Remelting process did not bring about a great change in thickness of each deposited layer. Before and after remelting, the thickness of each layer was about 0.4 mm. The thicknesses of two ten-layer coatings were similar, both of which were 3.8 mm.”
Point 2: Figure 2 is an XRD diffraction pattern of the LFRed K417G superalloy coating.
XRD measurements can detect crystal structure of a phase but typically not the composition of the phases. However, in Fig. 2 the authors describe the second phase as Ni-Cr-Co-Mo. In the description in the corresponding text they mention it is the matrix g phase. In Table 2 they also present quantitative measurement of composition from different points in the microstructure and this includes all the elements (C, Al, Ti, Cr, Fe, Ni, Mo, Co) for all points measured by EDS. So it is inadequately described in the paper how the authors concluded that the g phase has only Ni, Cr, Co and Mo. Also it is more appropriate to mark the second phase in Fig 2 as g phase and not Ni-Cr-Co-Mo.
Response 2: I made a mistake here, that is, I just marked the second phase in Figure 2 by the name of the PDF card. I'm sorry for my carelessness. You are right. The matrix γ phase includes all the elements (C, Al, Ti, Cr, Fe, Ni, Mo, Co), so I should not describe the γ phase as Ni-Cr-Co-Mo.
According to your suggestion, I have marked the second phase in Figure 2 as γ phase.
Point 3: In order to further analyze the microstructure and phase composition, …. dark-gray zone (point 1 in Figure 4b) is g matrix phase. (Lines 136-143)
It is concluded here that the EDS measurement at point 1 is from the g matrix. Which means that the microstructure in the dendrite is single phase g matrix? This is surprising as I would expect a two phase g + g’ microstructure. To best resolve this, the authors should present a higher magnification SEM image from this area.
In general, the microstructure images presented in the paper are not always of the right quality / magnification to follow the descriptions of the structure provided. This must be improved.
Response 3: Yes, the microstructure in the dendrite is a two phase γ + γ′ microstructure. I have presented a higher magnification SEM image of the dendrite. In addition, I have presented new SEM images of the as-deposited microstructure and revised the description of the relevant paragraph.
The new SEM images are in the following new Figure 4, while the higher magnification SEM image of the dendrite is Figure 4g.
The relevant description I revised is that “As can be seen from Figure 4g, there are also dot-like precipitates in the dark-gray zone, which are the γ' phases precipitated on the matrix. Therefore, the dark-gray zone (point 1 in Figure 4b) is a two phase γ plus γ' microstructure.”
Point 4: In addition, the cracks are always ….. presented by Carter et al [27]. (Lines 165-169)
It is not clear why crack count and crack density is more in X-Y section than in the other sections? (see Fig. 5b). The authors should comment on this.
Also in Fig 5a, the original HAZ microstructure is hardly visible and almost totally covered by the annotation used. Microstructure of both the HAZ and the interface between HAZ and RZ are very important for readers to follow the results presented in this paper. Therefore, suitable change in Fig. 5a is necessary to take care of this shortcoming. A new image with larger area from the HAZ will be even better.
Response 4: Here, I failed to point out what I mean very well. It is that the cracks on the X-Z section are always approximately parallel to the depositional direction, as shown in Figure 5a. This is just looking at the direction of the cracks on the X-Z section.
The crack count and crack density (in Figure 5b) is a statistical result. I selected five SEM images with the same magnification at different positions in each section. Then I observed and calculated the crack count and crack density in each section. To be honest, I am not clear why the crack count and crack density is more on the X-Y section than on the other sections. I presume it is probably because of the anisotropy. In fact, I presented the crack count and crack density on the each section with the purpose of more comprehensively expressing the serious cracking behavior in the as-deposited K417G. This result is also important to the discussion of the effect of laser remelting process on the cracking behavior.
Anyways, I think the reason why the crack count and crack density is more on the X-Y section than on the other sections can be a research direction for me in the future. So thank you again.
According to your suggestion, I have presented a new image with larger area from the HAZ.
Point 5: Consequently, a large amount of g' phase exists in the K417G superalloy. (Line 179)
What is the g’ volume fraction? An estimate can be easily obtained from the XRD results.
Response 5: The formula I found for measuring the phase content using XRD requires the RIR value of PDF cards for all phases. Unfortunately, the RIR values in the PDF cards of these phases are missing, so I cannot obtain the γ′ volume fraction from the XRD results as you requested.
Anyways, I used metallographic methods to measure the γ′ volume fraction. The specific steps are as follows.
First, I selected five SEM images of the microstructure with a magnification of 700 times. Image-pro Plus 6.0 analysis software was used to calculate the dendritic area fraction and the interdendritic area fraction, which was about 42% and 58%, respectively.
Next, I selected five SEM images of the microstructure in the dendrites and between the dendrites with a magnification of 10000 times respectively. I removed the areas of the carbides and the γ + γ′ eutectic. Then I calculated the area fraction of the γ′ phase in the dendrites and between the dendrites, which was about 23% and 89%, respectively.
Therefore, the γ′ phase area fraction is about 61%, so I think we could approximate the γ′ phase volume fraction to be 61%. This is an estimated value.
In fact, as K417G is a superalloy of the Chinese brand, it has been mentioned in the Chinese superalloy manual that the γ′ volume fraction in cast K417G is greater than 67%. I presume the 6% difference is caused by the error in my statistics on the one hand, and on the other hand, the rapid cooling of the laser deposition process may prevent the γ′ phase from fully precipitating. Therefore, I believe that if heat treatment has been taken place on the as-deposited K417G, more γ′ will be obtained, which can play a more significant role in strengthening. Thank you for your inspiration.
In addition, perhaps because of my limited knowledge, it is difficult for me to calculate the γ′ volume fraction by XRD, so I can only use the metallographic method mentioned above. I will continue to study the analysis software in depth. If it is convenient, I hope you can send me the method of calculating phase content by XRD or relevant literature via my email. Thank you again.
Point 6: Secondly, the processing characteristic of LFR ….. cladding takes place. (Lines 183-189)
Microstructural study can easily reveal details about the different layers and their thicknesses. It is very important information for this study, but is totally missing. For any understanding of the cracking behavior these microstructural details are absolutely essential.
Response 6: According to your suggestion, I have added an image to Figure 6 as Figure 6a to help the readers understand the details about the different layers and their thicknesses, and better understand the five thermal cycles described in Figure 6b.
Point 7: Fig. 6. Some details are not clear.
a. In this diagram - Left to Right is cooling, i.e. temperature decreases (I presume). Such details must be explicitly indicated in the diagram.
b. It must also be mentioned - under what conditions / stages, the different Thermal Cycles (1-5 marked) actually occur during the LFR processing.
Moreover, I fail to understand why both the heating and cooling parts of the Thermal cycles (1-5) are included in the diagram and also why these thermal cycles are symmetrical in heating and cooling? I understand that in the LFR process power is supplied to the system to heat up (and thus is controlled) while during cooling the power is switched off (not controlled), therefore the cycle should be asymmetrical logically.
Response 7:
a. Yes, from left to right is the cooling process. As per your requirements, I have pointed it out in the diagram.
b. These heat cycles are also schematic. I have added some explanations of these five thermal cycles, along with Figure 6a, to give the reader a better understanding of how these thermal cycles occur.
You are right. I have also modified these thermal cycles to be asymmetrical according to your requirements, and I think the cooling speed will be slower than the heating speed.
Point 8: As shown in Figure 6, the ductility of the material changes as the temperature decreases. (Line 197)
What is the basis of this schematic – i.e. is the information based on experimental results or calculations? Also how was ductility measured / calculated at high temperatures?
What are the thermal cycles 1 to 5? Some descriptions are presented in text later, but not fully. It is better to describe the different cycles in the caption of Fig. 6. Also it is better to define the various temperatures (TL, Ts, Tg’, etc.) used in the diagram.
Response 8:
From your comments, I understand that the information I gave about Figure 6 is vague, and I think there are mainly three problems I failed to explain clearly.
Firstly, the ductility curve is a schematic curve of high-temperature ductility of austenitic materials which I drew based on the research results of the high-temperature ductility of metals by Rhines F N. and Wray P J.
- Rhines, F.N.; Wray, P.J. Investigation of the intermediate temperature ductility minimum in metals. Transactions of the ASM. 1961, 54, 117-128.
Jingjing Yang also used this schematic curve as an assist to explain the cracking mechanism of RENE104 during direct laser fabrication.
- Yang, J.J.; Li, F.Z.; Wang, Z.M.; Zeng, X.Y. Cracking behavior and control of Rene 104 superalloy produced by direct laser fabrication. J. Mater. Process. Technol. 2015, 225, 229–239.
I think this schematic curve is reliable and shows the ductility changes during the solidification of the superalloy.
Second, the difference between the five thermal cycles is the different peak temperature of them. LFR is not a process in which the material as a whole heats up and cools down. Instead, the material at different locations will undergo melting and cooling successively on a certain path.
In the process of LFR, an already solidified position (except the final melting zone at the top) will undergo many thermal cycles when the laser is focusing on other position. And the peak temperature of the thermal cycle changes continuously along with the distance from the already solidified position to the molten pool. When the distance is close (for example, the next layer), the reheating temperature of the previous position will be high, while when the molten pool is further away, the reheating temperature will be low. As for the region A in Figure 6a, when the molten pool is in region A, the thermal cycle 1 occurs, while when the molten pool is in region B, the thermal cycle 2 occurs, and so on. These five cycles correspond to exactly five different distance ranges.
It is worth noting that the different regions shown in Figure 6a are distributed in three dimensions. In addition, the region A, B, C, D and E are also schematic and should actually be much larger than that in Figure 6a. I have added these contents in section 3.2.1.
Finally, the temperature of TL, TS, TE and Tγ' is the liquidus temperature, the solidus temperature, the eutectic reaction temperature, and the γ' precipitation temperature, respectively. I forgot to define Tγ' in the paper before. I am sorry for that.
In addition, it is difficult for me to measure these temperatures accurately. I originally intended to use the Ni-Al binary phase diagram to give an estimate, but considering that K417G contains too many alloying elements, the estimation error will be large. Therefore, I can only point out in Figure 6b that TL to Tγ' is a cooling process as you mentioned before.
Point 9: …the cracking behavior will be generated. ….. ductility dip cracking in LFRed K417G superalloy. (Lines 207-209)
The whole basis of the conclusion is only a schematic diagram - for which no information is provided - regarding how it was constructed.
Response 9: I have confused the causality here. Figure 6 is only used to assist in describing the cracking mechanisms. But the cracking mechanisms cannot be inferred from Figure 6. Therefore, I have removed the words "therefore, infer" and revised the sentence to "The cracking behavior in K417G during LFR could be classified into three types. They are solidification cracking, liquation cracking and ductility dip cracking."
Point 10: When the temperature drops below TS, these different orientations of solids grow alternately to form the dendrite skeleton. (Lines 214-215)
Schematic in Fig. 7 shows only a grain structure and there is no correlation revealed with respect to the dendrite structure. On the other hand, the dendrite structure is discussed at length in relation to LFR and cracking.
Response 10: In my understanding, in as-deposited structure, a grain consists of many dendrites with the same orientation. The interdendritic zone can be inside the grain or at the grain boundary. When the eutectic structure forms, it is easy to form the shrinkage cavity and microcrack in the weak eutectic structure between the dendrites. Because of the relatively high bonding strength within the grains, these defects usually do not expand into cracks. However, as the grain boundary strength is relatively low, these defects at the grain boundary will expand along the grain boundary and form cracks.
According to your suggestion, most of the descriptions have been modified to make them more consistent with Figure 7.
Point 11: Therefore, most of solidification cracks will…the dendrites remain free in the liquid phase and appear as round grains with smooth surface. (Lines 237-242)
Many things are not clear from the image or the description:
1. c & d are the images of the crack surface of the crack shown in a & b?
2. The crack path in a is tortuous and so why is the surface flat?
3. The description discusses denritic and interdendritic regions - which are these regions? Should be maked in image explicitly.
4. Discussion about liquid phase and round grains is not clear.
Response 11:
1. Yes, Figure 8c and d are the images of the crack surface of the crack shown in Figure 8a and b. Due to carelessness, I forgot to describe the crack shown in Figure 8a and b in the discussion, and I have added the description.
2. I think this may be caused by two reasons. One is that the magnification of two images (Figure 8a and c) is different; the other is that after the sample is broken apart, it is difficult to determine where the region (Figure 8a) is in the Figure 8c. It's a very long, thin crack, and it's actually straight at the macro level. I chose the images with larger magnification in order to look for the eutectic with shrinkage cavity and microcrack.
3. According to your suggestion, I have maked the denritic and interdendritic regions in Figure 8a.
4. I think that the dendritic growth near the crack is not hindered by the solid phase, and therefore the dendrite retains the smooth surface morphology which is in the liquid phase.
So the discussion about liquid phase and round grains I revised is that “Because the dendrites near the grain boundary remain free in the residual liquid phase and grow without solid obstacle, and the cracking is carried out along the grain boundary, the dendrites near the crack appear as round particles with smooth surface, as shown in Figure 8d.”
Point 12: In other words, there are countless HAZ inside the coating as long as the laser melting is not performed only once. (Lines 255-256)
This is a nice description to show that in LFR process HAZ is also formed layer wise. What I fail to find in this paper is a nice microstructural image to show this point - firstly and then secondly relate the position of the liquation cracks in these layers.
How thick are these deposition layers. It is a pitty, the paper lacks all these basic information - which is very important for such a study.
Response 12: Thank you for your praise of this description. I have added a description of the HAZ in combination with the microstructure shown in Figure 6a. The description is that “In Figure 6a, for example, when the molten pool is in the region C, the solidified region A will be a HAZ. While when the molten pool is in the region D, the solidified region B will be a HAZ at this time.”
In addition, as can be seen from the description in section 2 and Figure 6a, the thickness of each deposition layer is about 0.4mm. However, I think it is difficult to accurately mark where the HAZs are in Figure 6a. In the process of multi-layer laser forming, the position of laser focus changes constantly in three-dimensional space. Therefore, a certain position that has been solidified (for example, the region A in Figure 6a) will undergo many thermal cycles of different peak temperatures with the continuous change of distance from the molten pool (in the region B, C, D and E in Figure 6a). When the laser focuses on the region C (in Figure 6a), the solidified region A will be the HAZ. While when the laser focuses on the region D, the solidified region B will be the HAZ at this time. So I think HAZ is possible everywhere in the LFR coating.
Point 13: Once the liquefied microcrack is formed at the grain boundary, it will expand along the grain boundary with the accumulation of residual tensile stress and form the macroscopic crack through many deposited layers. (Lines 261-263)
Are we discussing here HAZ boundaries or grain boundaries?
Response 13: Here, I am discussing the grain boundaries. I think when the reheating temperature is above the melting point of the eutectics in HAZ, these eutectics will be remelted and form liquefied microcrack. However, because of the strong binding force inside the solidified grains, only the liquefied microcrack formed at the grain boundary can extend.
Point 14: The crack propagation requires more energy to be prevented, unless the steering dendrite with greater directional variation or the equiaxed grain is encountered. (Lines 264-265)
Not at all clear what is meant.
Response 14: Here I tried to explain how the crack extension stops, but now I find it is difficult to fully explain it based on experimental results and my current knowledge, and it has little to do with the discussion of cracking mechanisms. So I have deleted this confused description.
Point 15: Liquation cracks usually grow along the direction of dendrite growth, which is similar to an upward growth direction parallel to deposition. (Lines 268-269)
Bit confusing - isn't the deposition layers parallel to the substrate? Then of course the HAZ layer also should be parallel to the substrate. However, the growth of the dendrite then should be perpendicular to the substrate.
Or am I totally confused - in that case the authors must describe it in a better way for the readers to follow.
Response 15: At the last of this sentence, I want to write “the deposition direction” or “the cladding direction”. However, I carelessly omitted the word “direction”. I'm sorry about that. Yes, the growth direction of the dendrite is perpendicular to the substrate. That is what I meant to say.
Therefore, I revised this sentence to that “Liquation cracks usually grow along the direction of dendrite growth, which is similar to an upward growth direction parallel to the deposition direction.”
I have realized that the omission of a word can cause a great ambiguity. Sorry again.
Point 16: the grain boundary surface is rather round with dendritic protrusions, and obvious liquation of dendrite protrusions can be observed. (Lines 273-274)
Which is the grain boundary surface and which are the dendritic protrusions in Fig 10d - please mark in the figure.
Response 16: After second thought, I think it is not appropriate to describe "grain boundary surface" here. It cannot be seen from Figure 10d that where the grain boundary surface is, much less it is round. So I revised the description here as that “As shown in Figure 10d, obvious liquation of dendrite protrusions can be observed on the microscopic morphology of the crack section.”
In addition, I have marked the dendritic protrusion in Figure 10d.
Point 17: When the material is in during the thermal cycle 4 (the blue line Figure 6b), it will suffer the effect of continuously growing solid-phase shrinkage stress. (Lines 286-287)
Under what condition thermal cycle 4 occur?
Response 17: The thermal cycle 4 will occur when the solidified material has been reheated to the temperature range of Tγ' to TE and then rapidly cool down. In other words, when the molten pool is in the region D (Figure 6a), the material in the region D will experience thermal cycle 4 at this time. I realized that I had given too little information about the five thermal cycles before. I think the Figure 6a and related description I added in section 3.2.2 can help readers better understand the five thermal cycles in Figure 6b.
Point 18: When the process of diffusion or climb meets obstacles, it will lead to cracking due to strain concentration. (Lines 289-290)
1. From this description - every polycrystalline Ni-superalloy will develop crack during secondary creep - which is not true.
2. The process of diffusion or climb do not meet any obstacle, but the disloctions only meets obstacle when these processes are occuring.
Response 18:
1. Thank you for your correction. I think it depends on the inherent ductility of the material. K417G is a superalloy with high strength and poor ductility, so cracking is more likely to occur.
2. You are right. I'm confusing some concepts here. Thank you again for your correction.
So I revised the description here as that “When the processes of diffusion and climb are occurring, the dislocation will meet obstacles. And if the ductility of the material is poor, cracking will occur due to the strain concentration.”
Point 19: In fact, the precipitates play a double role in the effect on the sensitivity of ductility dip cracking. (Lines 296-297)
Presumably, here the precipitates are the carbides and not gamma prime. So it is better to be explicit and mention which precipitate?
Response 19: Thank you for pointing it out, here the precipitate is the carbide and not the γ'. According to your comment, I have revised the description of the precipitate to carbide.
Point 20:
What is the deposition on the substrate in this case?
Presuemably, it is the LFR coating on K417G substrate - so the figure caption should clearly mention this - in order not to create any confusion.
Response 20: Yes, it is the LFRed coating on K417G substrate.
According to your comment, I have revised the annotation of Figure 13 as that “Surface morphology of the LFRed coating on K417G substrate (a) before and (b) after laser remelting.”
The words in the end: I have tried my best to respond to your comments and revise the paper according to your suggestions. And in this process, I gained a lot of knowledge and inspirations. When I graduate, I intend to do postdoctoral research. I wonder if I can have the opportunity to learn from you. I know my research is not deep enough, sometimes I will make mistakes, and my English level is not high enough. But I will study harder to overcome these shortcomings. Anyway, once again, thank you very much.
Sincerely Yours,
Shuai Liu
E-mail: 200804892 @qq.com
Reviewer 2 Report
In this manuscript, the authors studied the Cracking, Microstructure and Tribological Properties of Laser Formed and Remelted K417G Ni-based Superalloy. The paper in general is interesting and well written. The only comment is that the authors need to link their findings with previously published work; the below are example papers for which the authors need to cite and link it to their work:
- Scanning strategies for texture and anisotropy tailoring during selective laser melting of TiC/316L stainless steel nanocomposites
- Thermal behavior of the molten pool, microstructural evolution, and tribological performance during selective laser melting of TiC/316L stainless steel nanocomposites: Experimental and simulation methods
- Nickel-based superalloys for advanced turbine engines: chemistry, microstructure and properties
- Evaluation of the mechanical properties of Inconel 718 components built by laser cladding
Author Response
Point: In this manuscript, the authors studied the Cracking, Microstructure and Tribological Properties of Laser Formed and Remelted K417G Ni-based Superalloy. The paper in general is interesting and well written. The only comment is that the authors need to link their findings with previously published work; the below are example papers for which the authors need to cite and link it to their work:
- Scanning strategies for texture and anisotropy tailoring during selective laser melting of TiC/316L stainless steel nanocomposites
- Thermal behavior of the molten pool, microstructural evolution, and tribological performance during selective laser melting of TiC/316L stainless steel nanocomposites: Experimental and simulation methods
- Nickel-based superalloys for advanced turbine engines: chemistry, microstructure and properties
- Evaluation of the mechanical properties of Inconel 718 components built by laser cladding
Response : Thank you very much for your recognition of our work.
I have read the papers you recommended to me. From the second paper and the third paper, I learned about the influence of process factors and composition factors on material properties, respectively. The first paper and the fourth paper helped me optimize the scanning path in the laser forming process. These papers are closely related to my work and give me great inspiration. Thank you again.
According to your suggestion, I have cited and linked these papers to my work in reference 5, 21-23.